# Empirical Evaluation of Knowledge Distillation from Transformers to Subquadratic Language Models

## Abstract

Knowledge distillation is a widely used technique for compressing large language models (LLMs), in which a smaller student model is trained to mimic a larger teacher model. Typically, both the teacher and student models are Transformer-based architectures, leveraging softmax attention for sequence modeling. However, the quadratic complexity of self-attention during inference remains a significant bottleneck, motivating the exploration of subquadratic alternatives such as structured state-space models (SSMs), linear attention, and recurrent architectures. In this work, we systematically evaluate the transferability of knowledge distillation from a Transformer teacher model to eight subquadratic student architectures. Our study investigates which subquadratic model can most effectively approximate the teacher model's learned representations through knowledge distillation, and how different architectural design choices influence the training dynamics. We further investigate the impact of initialization strategies, such as matrix mixing and query-key-value (QKV) copying, on the adaptation process. Our empirical results on multiple NLP benchmarks provide insights into the trade-offs between efficiency and performance, highlighting key factors for successful knowledge transfer to subquadratic architectures.

## 1 Introduction

The Transformer architecture (Vaswani et al., 2017) has led to significant advances in natural language processing (NLP) by enabling highly scalable and parallelizable training of language models (LMs). The core of its effectiveness is the self-attention mechanism, which produces contextualized token representations across long sequences. However, the quadratic computational complexity of self-attention, $\mathcal{O}(n^2)$ with respect to sequence length, leads to high inference costs for long sequences, posing challenges for resource-constrained applications.

**Rise of linear complexity architectures.** To address this limitation, alternative architectures have been proposed that reduce the complexity of self-attention. These models achieve subquadratic, and often linear, complexity with $\mathcal{O}(n)$. These include linear attention models (Katharopoulos et al., 2020), structured state-space models (SSMs) (Gu & Dao, 2024; Dao & Gu, 2024), and recurrent neural networks (RNNs) with improved gating mechanisms (Sun et al., 2023). These architectures aim to reduce computational overhead while maintaining competitive modeling capabilities.

While these architectures offer theoretical efficiency gains, pretraining them from scratch is prohibitively expensive and training-intensive. Moreover, their training dynamics remain less well understood than those of Transformers, making optimization more challenging. To avoid costly pretraining, we apply knowledge distillation (Hinton et al., 2015) from capable Transformer models into subquadratic architectures, aiming to retain their language modeling capabilities while significantly improving efficiency. Although knowledge distillation is typically applied between models of the same architecture, we adapt this paradigm to distill from a Transformer teacher into various subquadratic student models.

**Contributions.** To assess the feasibility of transferring knowledge from Transformer-based models into subquadratic architectures, we conduct a controlled empirical study involving eight distinct architectures (see Figure 1 for an overview of our approach). Our study aims to quantify the extent to

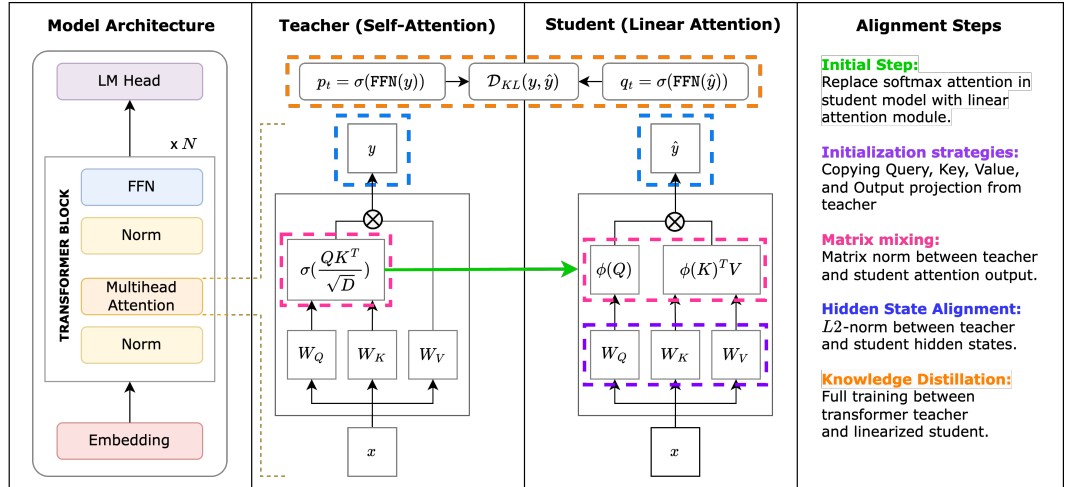

Figure 1: Overview of our knowledge distillation approach. We replace the softmax attention mechanism in transformer models with various subquadratic modules and train the resulting models using knowledge distillation and additional alignment techniques.

which different architectures preserve the inductive biases and representations learned by attention-based Transformers, and to analyze the effect of various alignment strategies on downstream task performance.

Specifically, we incorporate several alignment strategies to facilitate effective knowledge transfer, including *matrix mixing* (aligning the student's attention mechanism with the teacher's self-attention), *QKV copying* (initializing the student's query, key, and value projections with those learned by the teacher), and *hidden-state alignment* (minimizing the divergence between intermediate representations of the student and teacher models).

Our empirical results reveal significant performance disparities across different subquadratic architectures, with xLSTM Beck et al. (2024) achieving the highest average performance. Additionally, leveraging all advanced alignment techniques combined yields notable improvements. We summarize our contributions as follows:

- We present a systematic empirical evaluation of knowledge distillation into subquadratic models, comparing alignment techniques and downstream task performance.

- We study the effect of QKV copying, matrix mixing and hidden-to-hidden alignment, finding hidden-state alignment most consistently improves transfer.

- We introduce diagnostics of head diversity and teacher–student overlap, revealing which architectures best approximate attention allocation.

- We release our code and models to facilitate further research on linearizing attention-based Transformer models.

## 2 PRELIMINARIES AND RELATED WORK

With the introduction of Transformers (Vaswani et al., 2017), the softmax attention mechanism became the *de facto* standard for language modeling. However, it has a computational complexity of $\mathcal{O}(n^2 d)$, where $n$ is the sequence length and $d$ the hidden dimension of the model.

**Parallel form of softmax attention.** Given an input sequence $\boldsymbol{x} \in \mathbb{R}^{n \times d}$, the model computes projected "query," "key," and "value" representations as $\boldsymbol{Q}, \boldsymbol{K}, \boldsymbol{V} = \boldsymbol{x}\boldsymbol{W}_Q, \boldsymbol{x}\boldsymbol{W}_K, \boldsymbol{x}\boldsymbol{W}_V$, where $\boldsymbol{W}_Q, \boldsymbol{W}_K, \boldsymbol{W}_V \in \mathbb{R}^{d \times d}$ are learnable weight matrices. The output $\boldsymbol{y} \in \mathbb{R}^{n \times d}$ of softmax attention is computed as:

$$\boldsymbol{y} = softmax((\boldsymbol{Q}\boldsymbol{K}^{\intercal}) \odot \boldsymbol{M})\boldsymbol{V}, \qquad (1)$$

where $\boldsymbol{M} \in \mathbb{R}^{n \times n}$ is a causal mask to prevent the model from attending to future tokens. Thus, softmax attention allows each token to attend to all tokens in the sequence by computing similarity scores between queries and keys, and using these scores to compute a weighted sum of value vectors.

| ARCHITECTURE | RECURRENCE | DECAY TERM |
|---|---|---|
| mLSTM (Beck et al., 2024) | $\boldsymbol{S}_t = f_t \boldsymbol{S}_{t-1} + i_t \boldsymbol{v}_t \boldsymbol{k}_t^\top$ | dynamic |
| GLA (Yang et al., 2024) | $\boldsymbol{S}_t = \boldsymbol{S}_{t-1} \mathrm{Diag}(\boldsymbol{\alpha}_t) + \boldsymbol{v}_t \boldsymbol{k}_t^\top$ | dynamic |
| RetNet Sun et al. (2023) | $\boldsymbol{S}_t = \gamma \boldsymbol{S}_{t-1} + \boldsymbol{v}_t \boldsymbol{k}_t^\top$ | static |
| MetaLA (Chou et al., 2024) | $\boldsymbol{S}_t = \boldsymbol{S}_{t-1} \mathrm{Diag}(\boldsymbol{\alpha}_t) + \boldsymbol{v}_t (1 - \alpha_t)^\top$ | dynamic |
| DeltaNet (Yang et al., 2025) | $\boldsymbol{S}_t = \boldsymbol{S}_{t-1}(\alpha(\mathrm{I} - \beta_t \boldsymbol{k}_t \boldsymbol{k}_t^\top)) + \beta \boldsymbol{v}_t \boldsymbol{k}_t^\top$ | dynamic |
| Linear Attention | $\boldsymbol{S}_t = \boldsymbol{S}_{t-1} + \boldsymbol{v}_t \phi(\boldsymbol{k}_t)^\top$ | - |
| + Vanilla Choromanski et al. (2022) | where $\phi(x) = elu(x) + 1$ | - |
| + ReBased (Aksenov et al., 2024) | where $\phi(x) = (\gamma \cdot norm(x) + \beta)^2$ | - |
| + Hedgehog (Zhang et al., 2024b) | where $\phi(x) = \exp(Wx + b)$ | - |

Table 1: Overview of all architectures and their recurrent form under evaluation. $\boldsymbol{S}_t \in \mathbb{R}^{d \times n}$

**Recurrent form for inference.** While self-attention can be computed in parallel during training (Equation (1)), which is efficient on GPUs, inference requires sequential computation. At each decoding step, a newly generated token $\boldsymbol{x}_t \in \mathbb{R}^{1 \times d}$ attends to all previous tokens. Thus, the recurrent formulation of softmax attention is given by

$$\boldsymbol{y}_t = \frac{\sum_{i=1}^{t} exp(\boldsymbol{q}_t \boldsymbol{k}_i^\top) \boldsymbol{v}_i}{\sum_{i=1}^{t} exp(\boldsymbol{q}_t \boldsymbol{k}_i^\top)}, \tag{2}$$

where $\boldsymbol{q}_t, \boldsymbol{k}_t, \boldsymbol{v}_t = \boldsymbol{x}_t \boldsymbol{W}_Q, \boldsymbol{x}_t \boldsymbol{W}_K, \boldsymbol{x}_t \boldsymbol{W}_V$. As a result, autoregressive inference incurs growing memory and computational costs, since each new token must recompute attention over a ever-expanding set of keys and values $\{\boldsymbol{k}_i, \boldsymbol{v}_i\}_{i=1}^{t-1}$.

**Linear complexity with kernelized feature maps.** Katharopoulos et al. (2020) introduce a kernel-based approximation of the softmax attention by applying a feature map $\phi(\cdot)$, such that:

$$softmax(\boldsymbol{Q}\boldsymbol{K}^\top) \approx \phi(\boldsymbol{Q})\phi(\boldsymbol{K})^\top. \tag{3}$$

Leveraging the associative property of matrix multiplication, we can rewrite the recurrent form of attention:

$$\boldsymbol{y}_t = \frac{\sum_{i=1}^{t} \phi(\boldsymbol{q}_t)\phi(\boldsymbol{k}_i)^\top \boldsymbol{v}_i}{\sum_{i=1}^{t} \phi(\boldsymbol{q}_t)\phi(\boldsymbol{k}_i)^\top} \tag{4}$$

$$= \frac{\phi(\boldsymbol{q}_t) \sum_{i=1}^{t} \phi(\boldsymbol{k}_i)^\top \boldsymbol{v}_i}{\phi(\boldsymbol{q}_t) \sum_{i=1}^{t} \phi(\boldsymbol{k}_i)^\top}. \tag{5}$$

Unlike the standard softmax formulation (cf. Equation (2)), which scales with $\mathcal{O}(n^2 d)$, the kernelized approximation (cf. Equation (5)) reduces the complexity to $\mathcal{O}(nd^2)$.

**Existing Linear Attention Models.** Several feature map strategies have been proposed to address issues such as negative attention weights and training instabilities. **TransNormer** (Qin et al., 2022) and Retention Networks (**RetNet**) (Sun et al., 2023) identify instabilities in the normalization term of linear attention and replace classical normalization with GroupNorm (Wu & He, 2018). **Re-Based** (Aksenov et al., 2024) introduces a learnable polynomial kernel that adapts during training, mitigating the limitations of fixed feature maps. Similarly, **Hedgehog** (Zhang et al., 2024b) extends this idea by learning feature maps using single-layer networks, which preserve low-entropy attention weights and enforce monotonicity of query-key dot products. **DeltaNet** (Yang et al., 2025) introduces a delta update rule designed to improve memory efficiency and recall.

Beyond kernel-based methods, recent work incorporates recurrent structures into linear attention models. This includes Linear Recurrent Unit (**LRU**) (Orvieto et al., 2023) and Receptance Weighted Key Value (**RWKV**) (Peng et al., 2023; 2024), which both model sequence information through gated recurrence. Several works explore alternative gating parameterizations to improve selective

information flow. Examples include Gated Linear Attention (**GLA**) (Yang et al., 2024), Hierarchically Gated Recurrent Neural Networks (**HGRN/HGRN2**) (Qin et al., 2023; 2024), **Griffin** (De et al., 2024), and **mLSTM** (Beck et al., 2024). **Mamba2** (Dao & Gu, 2024) proposes a variant of linear attention based on state-space models from control theory, where sequence dynamics are modeled using latent state variables. Other approaches, such as Meta Linear Attention (**MetaLA**) (Chou et al., 2024) and Zimerman et al. (2024), present unified theoretical frameworks that improve the approximation of softmax attention while reducing parameter redundancy.

**Linearizing softmax attention in pretrained LMs.** Rather than training linear models from scratch, several approaches (Kasai et al., 2021; Mao, 2022) replace softmax attention with linear attention blocks in pretrained Transformers and apply knowledge distillation (Hinton et al., 2015). More recent work refines this paradigm with increasingly targeted strategies. SUPRA (Mercat et al., 2024) introduces a scalable uptraining framework to convert pretrained Transformers into recurrent architectures. LoLCATs (Zhang et al., 2024a) combines low-rank adaptation (Hu et al., 2021) with attention transfer to efficiently approximate softmax attention.. MOHAWK (Bick et al., 2024) employs a staged distillation pipeline that progressively aligns the student with its Transformer teacher. Further extensions include Mamba-LLaMA (Wang et al., 2025), which applies progressive distillation with instruction tuning, and LIGER (Lan et al., 2025), which reuses Transformer weights to construct gating modules for a range of subquadratic models, incorporating sliding-window attention. Finally, Yueyu et al. (2025) linearize Qwen-2.5 using RWKV-7 blocks, combining hidden-state alignment with word-level distillation. As Mamba has already become a common target for such distillation efforts, we focus our analysis on alternative subquadratic architectures.

## 3 METHODOLOGY

The first step in linearizing softmax attention-based language models involves replacing the attention block with a linear attention module (see Table 1). The common approach for training such linearized language models is to apply knowledge distillation (KD) from a softmax attention-based teacher model to a student model, thereby avoiding the need for expensive pretraining. The student model is trained using two objectives: *(1)* cross-entropy loss for next-token prediction and *(2)* the Kullback-Leibler (KL) divergence between output distributions of the teacher and the student. The total distillation loss $\mathcal{L}_{KD}$ is defined as:

$$\mathcal{L}_{\mathrm{KD}} = \mathcal{L}_{\mathrm{CE}} + \lambda \cdot \mathcal{L}_{\mathrm{KL}}, \tag{6}$$

where $\mathcal{L}_{\mathrm{CE}}$ is the cross-entropy loss and $\mathcal{L}_{\mathrm{KL}}$ is the KL divergence loss. $\lambda$ is a scaling factor controlling the contribution of each term. The KL divergence loss is given by:

$$\mathcal{L}_{KL} = \frac{1}{N} \sum_{i=1}^{N} \mathrm{KL}\big(p_T^{(i)} \| p_S^{(i)}\big), \tag{7}$$

where $N$ is the number of tokens, KL denotes the Kullback-Leibler divergence, and $p_T^{(i)}$ and $p_S^{(i)}$ are the output probability distributions of the teacher and student models, respectively, for the $i$-th token. We provide a conceptual overview of these two steps in Figure 1 and introduce additional alignment techniques in the following sections. As a preliminary verification, we confirm that knowledge distillation significantly improves student model performance and that parameter copying (e.g., copying the teacher's MLP layers, embeddings, and language modeling head) provides an effective starting point, consistent with prior findings (Appendix A).

### 3.1 ADDITIONAL ALIGNMENT IMPROVEMENTS

In the following section, we present refined alignment techniques to improve the distillation process between the transformer teacher model and the linearized student.

**Attention matrix alignment.** This approach aims to align the teacher's self-attention matrix with that of the linearized student model. However, this is non-trivial, since linear attention models do not explicitly compute full attention matrices. Prior work reconstructs approximate attention matrices from linear counterparts to enable alignment (Zhang et al., 2024b;a). In particular, the MOHAWK

framework (Bick et al., 2024) proposes a method based on minimizing the Frobenius norm between the teacher's self-attention matrix and the student's materialized matrix at each layer, referred to as "matrix mixing."

We extend this approach empirically to all eight linear architectures listed in Table 1. The matrix mixing loss is defined as:

$$\mathcal{L}_{\text{MM}} = \frac{1}{L} \sum_{i=1}^{L} \|\text{AttnMat}_{\text{T}}^{(i)} - \text{AttnMat}_{\text{S}}^{(i)}\|_F, \tag{8}$$

where $L$ is the number of layers, $\text{AttnMat}_{\text{T}}^{(i)}$ is the teacher's self-attention matrix at layer $i$, and $\text{AttnMat}_{\text{S}}^{(i)}$ is the materialized attention matrix of the student at the corresponding layer.

**Hidden state alignment.** An additional alignment strategy introduced in the MOHAWK framework is hidden state alignment, which encourages the student model's hidden representations to remain close to those of the teacher. This is achieved by minimizing the $L_2$-norm between corresponding hidden states at each layer. The hidden state alignment loss is defined as:

$$\mathcal{L}_{\text{H2H}} = \frac{1}{L} \sum_{i=1}^{L} \|h_T^{(i)} - h_S^{(i)}\|_2^2, \tag{9}$$

where $L$ is the number of layers, $h_T^{(i)}$ is the hidden state of the teacher model at layer $i$, and $h_S^{(i)}$ is the corresponding hidden state of the student model. This loss encourages the student model to preserve intermediate representations of the teacher, thereby improving structural alignment between the models.

## 4 EXPERIMENTAL SETUP

For our empirical evaluation, we consider eight subquadratic architectures as student models, listed in Table 1. We use SmolLM-360M (Allal et al., 2025) as our softmax attention-based teacher model, which is built on the Llama architecture (Touvron et al., 2023). To construct a linearized student model, we retain the teacher's normalization layers, MLP blocks, embedding layers, and language modeling head while replacing the self-attention mechanism with the corresponding linearized attention module (see Table 1). We show the exact parameter counts for each model in Appendix C.

We then train the student model using knowledge distillation, with additional alignment techniques progressively incorporated as described in Section 3. After training, we evaluate the student model's performance on various downstream tasks.

### 4.1 TRAINING DATASET AND EVALUATION

All student models are trained on a 3B-token subset of the FineWeb dataset (Penedo et al., 2024), a cleaned and deduplicated English web corpus. Text is concatenated and chunked into fixed-length sequences of 512 tokens. We allocate fixed budgets for alignment objectives: 80M tokens for matrix mixing and 160M for hidden-state alignment, following the MOHAWK setup (Bick et al., 2024). For evaluation, we follow LM-Eval-Harness (Gao et al., 2023) to assess six zero-shot tasks: LAMBADA (Paperno et al., 2016), WinoGrande (Sakaguchi et al., 2019), ARC (easy/challenge) (Clark et al., 2018), PIQA (Bisk et al., 2019), and HellaSwag (Zellers et al., 2019). LAMBADA is reported as the mean of its Standard and OpenAI variants. To evaluate long-context capabilities, we include five subsets from LongBench (Bai et al., 2024): WikiMQA, MultiFieldQA, NarrativeQA, TREC, and TriviaQA. Inputs exceeding the context window are left-truncated.

### 4.2 TRAINING DETAILS

We largely follow the training setup proposed in MOHAWK, using the Adam (Kingma & Ba, 2017) optimizer for matrix mixing, hidden state alignment and end-to-end training. For learning rate scheduling, we apply a stable decay schedule with warmup during matrix mixing phase and a

linear schedule for end-to-end training, which we found to yield more stable results across all model variants. The maximum learning rate was set to $1 \times 10^{-3}$, with a batch size of 48. We note that MOHAWK uses only the KL divergence as its final loss, whereas we additionally optimize with a cross-entropy loss term (see Equation (6)), as it is widely adopted in distillation setups and aligns with its use in many practical implementations (Sanh et al., 2019; Jiao et al., 2020; Haller et al., 2024). We primarily use FLA Yang & Zhang (2024) for model implementations, PyTorch Paszke et al. (2019) along with the Hugging Face Transformers and Datasets libraries Wolf et al. (2020); Lhoest et al. (2021) for model training, inference, and dataset management. We also compared the use of Frobenius norm vs. mean squared error (MSE) loss for matrix mixing and found both losses to perform similarly (Appendix A). Based on this observation, we opted for Frobenius norm alignment in our experiments due to its conceptual alignment with prior approaches (Bick et al., 2024).

## 5 EXPERIMENTS AND RESULTS

### 5.1 EXPERIMENT 1: DOWNSTREAM EVALUATION

Our first experiment aims to answer which subquadratic architectures are best suited for knowledge distillation from a Transformer-based teacher. To this end, we compare 8 architectures under different applications of the three phases of the MOHAWK framework: Stage 3 represents a full fine-tuning of the architecture and is always applied. Stages 1 and 2 correspond to attention matrix alignment and hidden state alignment, respectively. Applying all three phases constitutes to the full MOHAWK setup.

| MODEL | STAGES | LAMB. acc. | WINOG. acc. | ARC-E acc. norm. | ARC-C acc. norm. | PIQA acc. norm | HELLAS. acc. norm. | AVG.↑ | REC. |
|---|---|---|---|---|---|---|---|---|---|
| SmolLM-360M (Teacher) | - | 41.33 | 56.51 | 63.72 | 36.01 | 71.49 | 53.37 | 53.73 | - |
| Llama→Llama$_{fullcopy}$ | 3 | 40.88 | 56.04 | 63.01 | 36.35 | 71.44 | 53.59 | 53.55 | - |
| Llama→Llama$_{student}$ | 3 | 33.58 | 53.20 | 58.38 | 32.08 | 70.57 | 47.36 | 49.19 | - |
| Llama→Llama$_{student}$ | 2 + 3 | 40.75 | 56.99 | 63.43 | 36.26 | 71.60 | 53.10 | **53.68** | 99.90% |
| Llama→Llama$_{student}$ | 1 + 2 + 3 | 40.89 | 56.69 | 63.30 | 36.18 | 70.95 | 53.03 | 53.50 | - |
| Llama→xLSTM | 3 | 32.06 | 54.54 | 59.30 | 31.83 | 70.67 | 48.34 | 49.45 | - |
| Llama→xLSTM | 2 + 3 | 34.44 | 54.46 | 59.72 | 32.68 | 71.49 | 49.89 | 50.44 | - |
| Llama→xLSTM | 1 + 2 + 3 | 35.71 | 56.43 | 60.40 | 32.51 | 70.95 | 50.37 | **51.06** | 95.03% |
| Llama→MetaLA | 3 | 32.17 | 53.83 | 58.04 | 31.66 | 70.95 | 47.99 | 49.10 | - |
| Llama→MetaLA | 2 + 3 | 36.60 | 54.70 | 60.56 | 32.51 | 70.67 | 50.40 | 50.90 | - |
| Llama→MetaLA | 1 + 2 + 3 | 36.39 | 54.22 | 61.07 | 32.68 | 71.22 | 50.21 | **50.95** | 94.82% |
| Llama→GLA | 3 | 32.74 | 53.59 | 57.95 | 31.66 | 70.95 | 48.40 | 49.21 | - |
| Llama→GLA | 2 + 3 | 34.52 | 53.75 | 61.20 | 32.25 | 70.57 | 50.15 | 50.40 | - |
| Llama→GLA | 1 + 2 + 3 | 35.05 | 53.67 | 60.94 | 32.42 | 70.35 | 50.17 | **50.43** | 93.85% |
| Llama→RetNet | 3 | 30.01 | 53.04 | 57.41 | 32.17 | 69.86 | 46.45 | 48.15 | - |
| Llama→RetNet | 2 + 3 | 32.32 | 55.33 | 59.13 | 31.23 | 70.51 | 48.47 | **49.49** | 92.10% |
| Llama→RetNet | 1 + 2 + 3 | 31.54 | 53.83 | 59.97 | 32.00 | 70.35 | 48.47 | 49.35 | - |
| Llama→DeltaNet | 3 | 32.44 | 53.51 | 58.84 | 31.74 | 71.55 | 47.81 | 49.31 | - |
| Llama→DeltaNet | 2 + 3 | 28.28 | 52.49 | 57.32 | 31.74 | 70.46 | 46.38 | **47.77** | 88.90% |
| Llama→DeltaNet | 1 + 2 + 3 | 28.38 | 52.01 | 56.86 | 31.83 | 70.18 | 45.98 | 47.54 | - |
| Llama→VanillaLA | 3 | 19.03 | 50.20 | 51.01 | 27.65 | 67.68 | 38.53 | 42.53 | - |
| Llama→VanillaLA | 2 + 3 | 31.74 | 53.91 | 56.90 | 31.83 | 69.75 | 46.99 | **48.52** | 90.30% |
| Llama→VanillaLA | 1 + 2 + 3 | 30.94 | 53.75 | 55.68 | 31.48 | 70.02 | 46.33 | 48.03 | - |
| Llama→Rebased | 3 | 20.76 | 50.51 | 50.55 | 27.99 | 68.12 | 39.29 | 42.80 | - |
| Llama→Rebased | 2 + 3 | 31.77 | 53.35 | 58.25 | 30.97 | 69.80 | 47.60 | 48.62 | - |
| Llama→Rebased | 1 + 2 + 3 | 34.41 | 52.80 | 57.83 | 32.42 | 69.75 | 48.60 | **49.30** | 91.75% |
| Llama→Hedgehog | 3 | 20.57 | 51.07 | 52.06 | 28.58 | 68.66 | 39.43 | 43.95 | - |
| Llama→Hedgehog | 2 + 3 | 30.94 | 53.83 | 56.94 | 31.14 | 69.75 | 46.45 | **48.17** | 89.65% |
| Llama→Hedgehog | 1 + 2 + 3 | 30.72 | 53.99 | 56.99 | 30.38 | 70.57 | 46.18 | 48.13 | - |

Table 2: Results on Zero-Shot LM downstream benchmarks. All models, except the teacher model SmolLM-360M, were trained for 3B tokens of the FineWeb dataset. We provide two Llama-Llama results as upper bounds of transfer within the same architecture: (1) Llama→Llama$_{student}$, where a new transformer model is distilled from a teacher. (2) Llama→Llama$_{fullcopy}$, a sanity check where the teacher is fully copied into the student. We find that several subquadratic architectures, such as xLSTM and MetaLA, outperform the Llama→Llama$_{student}$ baseline.

As a point of reference, we include two configurations where the student is also based on the LLama architecture: one where a newly initialized LLama-based student is trained from the teacher (Llama→Llama$_{student}$) and a sanity check in which the full teacher model is copied into the student and then continuously fine-tuned (Llama→Llama$_{fullcopy}$). Table 2 shows the results of this comparison. We make the following observations:

**Recoverage of linearized models.** Among all student architectures, xLSTM, GLA, and MetaLA consistently achieve the highest recoverage scores across all training stage combinations, recovering up to 95% of the teacher model's performance. In contrast, models lacking dynamic decay mechanisms, like those with static or no decay terms, consistently underperform. This trend highlights the importance of explicit memory dynamics in preserving the inductive biases of the teacher during distillation.

**Subquadratic architectures without decay term consistently underperform.** Kernel-based attention models such as VanillaLA, Rebased, and Hedgehog fail to match the performance of recurrent or gated architectures, even when trained with advanced alignment strategies. Although Hedgehog incorporates learnable feature maps to approximate softmax attention, it does not outperform simpler baselines, indicating that capturing softmax-like properties alone is insufficient. These results highlight the importance of explicit memory mechanisms, such as decay or gating, for effectively transferring the teacher model's sequential reasoning capabilities.

**Hidden state alignment substantially boosts performance, especially on tasks requiring long-range reasoning.** We observe that hidden-state alignment and end-to-end training (Stages 2+3) yields consistent improvements across all architectures compared to full fine-tuning alone (Stage 3), with average gains of 1–3 points. These improvements are particularly pronounced on LAMBADA, a benchmark designed to test long-range dependency modeling. For example, MetaLA improves from 30.10 to 36.60 accuracy, and Rebased from 19.57 to 31.77.

**Attention matrix alignment only provides marginal improvements.** Extending training to include attention matrix alignment (Stages 1+2+3) provides only marginal improvements over hidden state alignment alone (Stages 2+3), and primarily for architectures that already provide a strong baseline. For most architectures, this phase has negligible or even negative impact, indicating that attention matrix alignment is only beneficial when the student model is structurally capable of representing softmax-style interactions.

For full details on the convergence behavior across training stages, we provide per-stage plots in Appendix D.

## 5.2 EXPERIMENT 2: IMPACT OF QKV COPYING

We conduct an ablation experiment to investigate whether copying the query, key, and value and output projections from the teacher model provides a good initialization for more effective alignment. To this end, we train each model both with and without copying all projections from the Transformer teacher. The results are shown in Table 3. We find that, while copying each projection offers a helpful initialization, it is insufficient for effective knowledge transfer on its own. Only for Llama→Hedgehog do we observe a noticeable improvement. This suggests that additional alignment stages are necessary to address structural mismatches and enable effective distillation.

| MODEL | W/O | +QKV | Δ |
|---|---|---|---|
| xLSTM | 49.45 | 49.19 | −0.26 |
| GLA | 49.21 | 49.10 | −0.11 |
| RetNet | 48.15 | 48.04 | −0.11 |
| DeltaNet | 49.31 | 46.64 | −2.67 |
| MetaLA | 49.10 | 48.56 | −0.54 |
| VanillaLA | 42.53 | 42.53 | ±0 |
| Rebased | 42.80 | 42.11 | −0.69 |
| Hedgehog | 43.95 | 44.87 | +0.92 |

Table 3: Effect of QKV copying on average downstream accuracy.

## 5.3 EXPERIMENT 3: LONG-CONTEXT EVALUATION

To assess the generalization ability of distilled models beyond standard sequence lengths, we evaluate them under long-context scenarios. First, we conduct controlled perplexity measurements on progressively longer input sequences to analyze each model's capacity to integrate and retain information over extended contexts. Second, we evaluate downstream performance using a subset of

tasks from the LongBench benchmark, which reflects realistic, context-heavy applications. For inputs exceeding a model's maximum context length, we apply left-truncation. As shown in Figure 2, models with dynamic decay terms, like xLSTM, GLA, and MetaLA, maintain stable performance across longer sequences. In contrast, models without such mechanisms (e.g., DeltaNet, RetNet, LA) exhibit significant degradation, indicating limited long-range generalization.

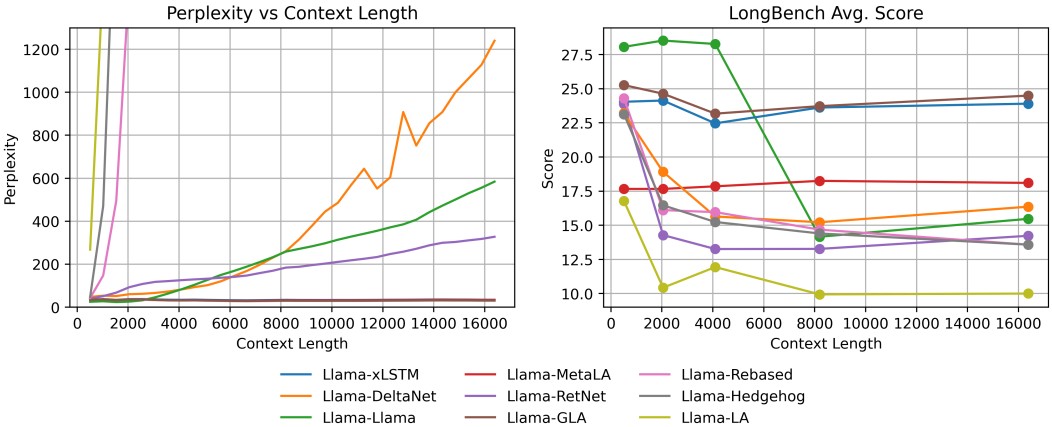

Figure 2: Long-context evaluation. Left: Perplexity over increasing context lengths. Right: Long-Bench scores. Models with dynamic decay terms (xLSTM, GLA, MetaLA) retain performance across increasing context lengths, while others show degradation.

## 5.4 EXPERIMENT 4: HEAD ANALYSIS

To complement downstream-based evaluation, we probe how student token mixers allocate mass compared to Transformer attention (Table 4). We report cosine similarity as a measure of head diversity and three alignment metrics with the teacher: Top-3 Jaccard (overlap of attended keys), mass on student (teacher mass on student Top-3), and mass on teacher (student mass on teacher Top-3).

| MODEL | STAGE | COSINE SIM↓ | JACCARD↑ | TOP-3 KEY ALIGNMENT MASS ON TEACHER | MASS ON STUDENT | AVG. |
|---|---|---|---|---|---|---|
| SmolLM-360M (Teacher) | – | 0.93 | 1.00 | 0.60 | 0.60 | 53.73 |
| Llama→xLSTM | 3 | 0.91 | 0.23 | 0.16 | 0.35 | 49.45 |
| Llama→xLSTM | 1+2+3 | 0.93 | 0.32 | 0.30 | 0.53 | 51.06 |
| Llama→MetaLA | 3 | 0.88 | 0.06 | 0.04 | 0.07 | 49.10 |
| Llama→MetaLA | 1+2+3 | 0.86 | 0.21 | 0.07 | 0.12 | 50.95 |
| Llama→GLA | 3 | 0.97 | 0.27 | 0.17 | 0.23 | 49.21 |
| Llama→GLA | 1+2+3 | 0.98 | 0.34 | 0.18 | 0.28 | 50.43 |
| Llama→Retnet | 3 | 0.50 | 0.40 | 0.10 | 0.12 | 48.15 |
| Llama→Retnet | 1+2+3 | 0.82 | 0.67 | 0.21 | 0.23 | 49.35 |
| Llama→Deltanet | 3 | 0.45 | 0.26 | 0.11 | 0.14 | 49.31 |
| Llama→Deltanet | 1+2+3 | 0.82 | 0.27 | 0.23 | 0.41 | 47.77 |
| Llama→LA | 3 | 0.91 | 0.16 | 0.08 | 0.23 | 42.53 |
| Llama→LA | 1+2+3 | 0.92 | 0.45 | 0.14 | 0.19 | 48.03 |
| Llama→Hedgehog | 3 | 0.62 | 0.01 | 0.04 | 0.04 | 42.80 |
| Llama→Hedgehog | 1+2+3 | 0.71 | 0.07 | 0.04 | 0.04 | 49.30 |
| Llama→Rebased | 3 | 0.40 | 0.02 | 0.06 | 0.05 | 43.95 |
| Llama→Rebased | 1+2+3 | 0.45 | 0.09 | 0.04 | 0.07 | 48.13 |

Table 4: Diagnostics for the same models as in Section 5.1. Cosine similarity quantifies head diversity, while Top-k Jaccard, mass on student, and mass on teacher capture how attention is allocated by the student and how it aligns with the teacher. The final column reports average downstream accuracy (reproduced from Section 5.1) for reference.

The results show that multi-stage alignment consistently improves teacher–student overlap. xLSTM and GLA benefit most, reaching higher Jaccard and mass values, while RetNet achieves the strongest alignment overall after staged training despite reduced diversity in single-stage. Deltanet improves in reciprocity but remains less consistent. By contrast, MetaLA, Hedgehog, and Rebased display persistently low alignment across both regimes, indicating that their allocation strategies diverge from the teacher even under stronger guidance. Overall, staged alignment appears most effective for architectures already predisposed to recall-like behavior (xLSTM, GLA, RetNet), while others resist convergence.

## 6 CONCLUSION

Our study evaluates the effectiveness of distilling Transformer-based language models into a range of subquadratic architectures, focusing on alignment techniques such as QKV copying, attention-, and hidden-to-hidden alignment. We find that models with dynamic decay mechanisms consistently achieve the highest performance and recover well across training stages. In contrast, models without explicit memory dynamics - such as VanillaLA, Rebased, and Hedgehog - struggle to match the teacher, even with advanced alignment strategies. While QKV copying serves as a convenient initialization, it is insufficient alone, highlighting the importance of progressive alignment.

Among the evaluated techniques, hidden-to-hidden alignment emerges as the most reliable strategy for guiding student models toward the teacher's representations. Attention alignment can further support this process, though its benefits are more architecture-dependent. Notably, several subquadratic models, such as xLSTM, GLA, and MetaLA, achieve strong downstream performance while preserving the efficiency advantages of linearized attention.

As an outlook, preliminary results with scaled variants of xLSTM (Table 9) suggest promising gains with increased model capacity. Future work may explore scaling and adapting hidden-state alignment for larger models.

We release our training pipelines, architectures, and evaluation framework to support continued research on efficient model design and cross-architecture distillation.

### LIMITATIONS

While our findings offer meaningful contributions, several limitations should be considered:

**Lack of qualitative analysis.** While we provide a broad empirical evaluation across diverse subquadratic backbones, we do not examine how the models' inductive biases manifest during the approximation of attention weights. A deeper analysis of the resulting attention patterns—e.g., spikiness, focus distribution, or alignment dynamics—could offer valuable insights into why certain architectures align better than others and inform future improvements to the distillation process.

**Limited training data.** The experiments were conducted with a constrained dataset, limiting our ability to assess the full generalization potential of the proposed techniques. Larger-scale training could reveal additional insights into model adaptation across diverse benchmarks.

**Scaling to larger models.** Our study primarily focuses on mid-sized models (350M to 500M parameters), and it remains an open question how well these techniques generalize to larger architectures. We hypothesize that matrix mixing may be more effective for larger models due to their increased hidden state dimensionality and greater representational capacity, allowing for a closer approximation of the teacher's attention matrix.

Despite these limitations, our findings provide a foundation for future work exploring more effective alignment techniques, improved compatibility layers, and novel training methodologies for efficient language models. Further research into alternative architectures and task-specific adaptations will be essential for advancing the deployment of subquadratic models in real-world applications.

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

## A  PRELIMINARY EXPERIMENTS

To validate our approach before full-scale training, we conducted preliminary experiments comparing standard training (without parameter copying) against parameter-initialized training on a next-token prediction task. Our goal was to assess whether initializing student models with parameters from a pre-trained Transformer teacher could provide a more effective starting point.

Additionally, we explored the effect of Frobenius norm vs. MSE loss for Attention Alignment, finding both to yield similar performance.

| MODEL | INITIALIZATION METHOD | LAMB. | WINOG. | ARC-E | ARC-C | PIQA | HELLAS. | AVG.↑ |
|---|---|---|---|---|---|---|---|---|
| SmolLM-360M | | 49.26 | 59.35 | 70.24 | 36.65 | 71.65 | 43.11 | 55.04 |
| *Preliminary Standard Training* | | | | | | | | |
| xLSTM | | 10.36 | 51.38 | 36.70 | 20.05 | 61.81 | 20.07 | 33.39 |
| Llama→xLSTM | | 22.09 | 53.20 | 52.03 | 25.09 | 67.95 | 35.36 | 42.62 |
| *Frobenius vs. MSE* | | | | | | | | |
| Llama→xLSTM$_{Frobenius}$ | *+ QKV + Matrix Mixing* | 34.13 | 55.17 | 66.40 | 29.01 | 70.62 | 38.54 | 48.98 |
| Llama→xLSTM$_{MSE}$ | *+ QKV + Matrix Mixing* | 33.76 | 55.41 | 65.43 | 29.35 | 70.24 | 38.55 | 48.79 |

Table 5: Preliminary experiments conducted on 1B tokens.

## B  ATTENTION MATRIX APPROXIMATION

Table 6 summarizes all models under evaluation and how each attention matrix equivalent is constructed. We furthermore include references to the original definition.

We define $\boldsymbol{CM}$ as the causal mask, where

$$\boldsymbol{CM}_{ij} = \begin{cases} 0, & \text{if } j \leq i \\ -\infty, & \text{if } j > i \end{cases} \tag{10}$$

| Architecture | Mixing Matrix $\boldsymbol{P}$ | Decay / Mask Term | Reference |
|---|---|---|---|
| Linear Attention | $\boldsymbol{P} = (\phi(\boldsymbol{Q})\phi(\boldsymbol{K})^\top) \odot \boldsymbol{CM}$ | | |
| + Vanilla | $\phi(x) = elu(x) + 1$ | - | |
| + Rebased | $\phi(x) = (\gamma \cdot norm(x) + \beta)^2$ | - | |
| + Hedgehog | $\phi(x) = \exp(Wx + b)$ | - | |
| GLA | $\boldsymbol{P} = ((\boldsymbol{Q} \odot \boldsymbol{B})(\frac{\boldsymbol{K}}{\boldsymbol{B}})^\top) \odot \boldsymbol{CM}$ | $\boldsymbol{B} = \prod_{j=i+1}^t \alpha_j^\top 1$ | Yang et al. (2024), Section 4.1 |
| mLSTM | $\boldsymbol{P} = \boldsymbol{Q}\boldsymbol{K}^\top \odot (\boldsymbol{F} \odot exp(\tilde{\boldsymbol{I}}))$ | $\boldsymbol{F}_{i,j} = \begin{cases} 0, & \text{if } i < j \\ 1, & \text{if } i = j \\ \prod \sigma(\tilde{f}_k), & \text{if } i > j \end{cases}$ | Beck et al. (2024), Appendix A.3 |
| RetentionNet | $\boldsymbol{P} = \boldsymbol{Q}\boldsymbol{K}^\top \odot \boldsymbol{D}$ | $\boldsymbol{D}_{i,j} = \begin{cases} 0, & \text{if } i < j \\ \gamma^{i-j} & \text{if } i \geq j \end{cases}$ | Sun et al. (2023), Section 2.1 Eq. 5 |
| DeltaNet | $\boldsymbol{P} = (\boldsymbol{Q}\boldsymbol{K}^\top \odot \boldsymbol{CM}) \odot \boldsymbol{T}$ | $T = (\boldsymbol{I} + tril(diag(\beta)\boldsymbol{K}\boldsymbol{K}^\top, -1))^{-1} \cdot diag(\beta)$ | Yang et al. (2025), Section 3.2 |

Table 6: Overview of attention matrix approximations for different sequence mixer backbones.

## C MODEL PARAMETER COUNTS

Table 7 lists the number of parameters for each model after replacing the attention layer with the corresponding linear attention backbone.

| Model | #Params |
|---|---|
| Llama | 361M |
| Llama→xLSTM | 478M |
| Llama→GLA | 478M |
| Llama→RetNet | 477M |
| Llama→MetaLA | 477M |
| Llama→DeltaNet | 448M |
| Llama→VanillaLA | 448M |
| Llama→Rebased | 448M |
| Llama→Hedgehog | 448M |

Table 7: Model list with corresponding parameter count

# D EXPERIMENT 1: CONVERGENCE BEHAVIOUR

Figure 3 provides an overview of loss trajectories across training stages for each model under all three stage configurations.

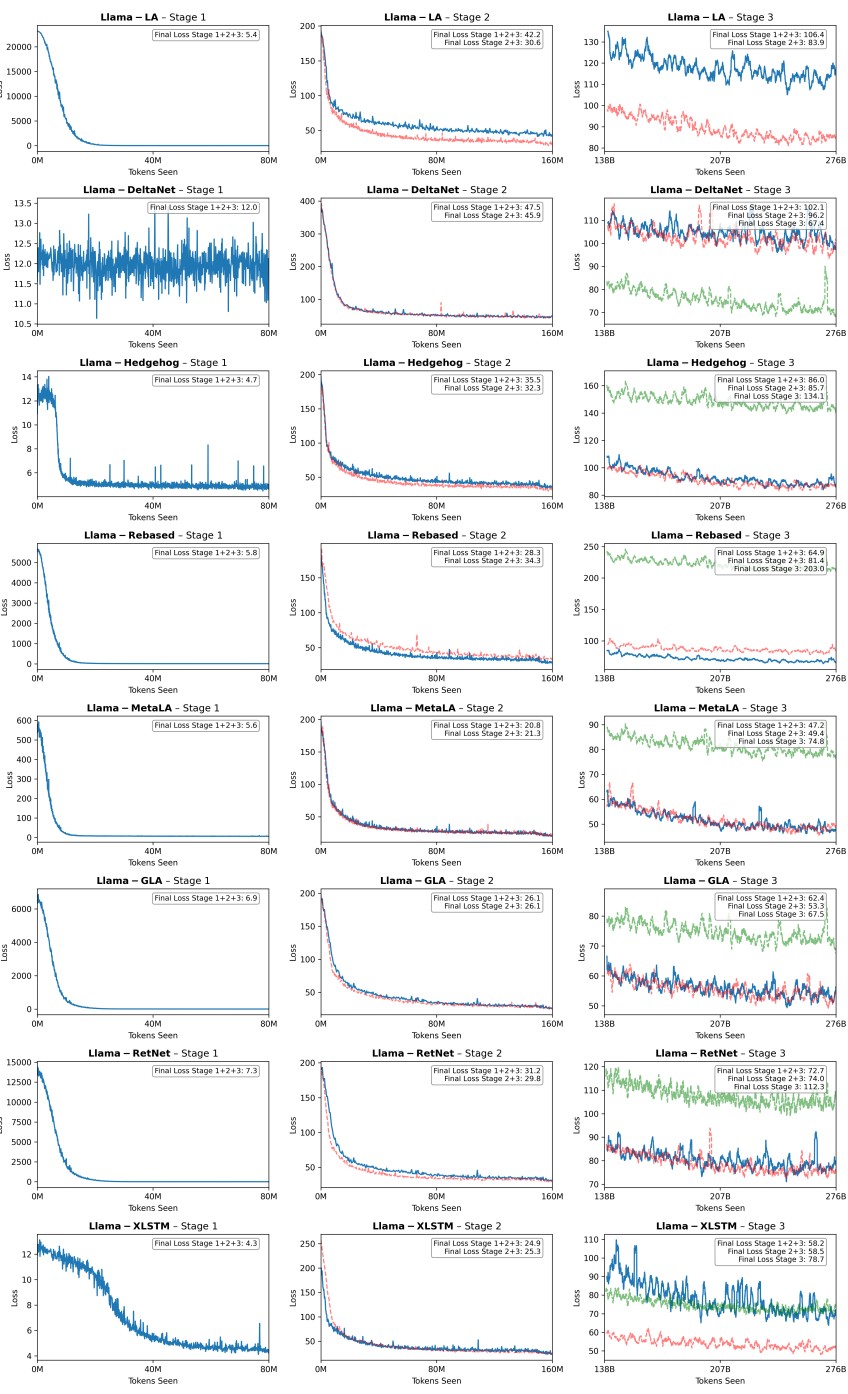

Figure 3: Loss plots for all runs conducted in Experiment 1. Green line plots indicate only Stage 3 training, while red and blue indicate Stage 2+3 and 1+2+3 Stage respectively.

# E   EXPERIMENT 3: FULL RESULTS FOR THE LONGE CONTEXT EXPERIMENTS

For completeness, we include the full results of Experiment 4.

| Model | WIKIMQA | MULTIFIELDQA | NARRATIVEQA | TREC | TRIVIAQA | AVG. |
|---|---|---|---|---|---|---|
| *512 Context* | | | | | | |
| SmolLM-360M | 34.30 | 26.71 | 30.25 | 14.96 | 34.11 | 28.06 |
| Llama→xLSTM | 31.90 | 23.94 | 26.54 | 7.67 | 30.15 | 24.04 |
| Llama→GLA | 34.12 | 29.26 | 28.92 | 5.75 | 28.26 | 25.26 |
| Llama→MetaLA | 22.59 | 21.19 | 19.46 | 0.00 | 25.04 | 17.66 |
| Llama→RetNet | 31.17 | 26.35 | 26.53 | 8.25 | 27.06 | 23.87 |
| Llama→DeltaNet | 26.19 | 27.38 | 27.44 | 5.25 | 29.79 | 23.21 |
| Llama→LA | 21.30 | 19.82 | 19.72 | 0.00 | 23.07 | 16.78 |
| Llama→Bebased | 32.61 | 28.54 | 24.78 | 9.00 | 27.51 | 24.49 |
| Llama→Hedgehog | 31.66 | 25.45 | 26.12 | 2.75 | 29.67 | 23.13 |
| *2K Context* | | | | | | |
| SmolLM-360M | 35.63 | 27.17 | 30.06 | 16.08 | 33.66 | 28.52 |
| Llama→xLSTM | 32.87 | 26.88 | 27.04 | 5.75 | 28.10 | 24.13 |
| Llama→GLA | 30.39 | 29.29 | 26.79 | 5.67 | 31.03 | 24.63 |
| Llama→MetaLA | 22.54 | 22.06 | 19.10 | 0.00 | 24.59 | 17.66 |
| Llama→RetNet | 18.00 | 17.40 | 16.03 | 1.50 | 18.48 | 14.28 |
| Llama→DeltaNet | 24.98 | 24.41 | 20.49 | 0.50 | 24.17 | 18.91 |
| Llama→LA | 11.75 | 11.36 | 12.99 | 0.00 | 16.06 | 10.43 |
| Llama→Rebased | 21.67 | 20.75 | 17.96 | 0.00 | 20.18 | 16.11 |
| Llama→Hedgehog | 22.28 | 20.02 | 21.13 | 0.00 | 18.88 | 16.46 |
| *4K Context* | | | | | | |
| SmolLM-360M | 33.18 | 24.51 | 31.70 | 15.29 | 36.68 | 28.27 |
| Llama→xLSTM | 31.16 | 23.40 | 25.77 | 5.00 | 26.96 | 22.46 |
| Llama→GLA | 33.12 | 23.05 | 26.83 | 2.75 | 30.10 | 23.17 |
| Llama→MetaLA | 22.73 | 22.71 | 19.10 | 0.00 | 24.73 | 17.85 |
| Llama→RetNet | 18.07 | 11.21 | 16.66 | 1.25 | 19.12 | 13.26 |
| Llama→DeltaNet | 16.71 | 18.49 | 19.55 | 0.00 | 23.44 | 15.64 |
| Llama→LA | 13.97 | 14.92 | 17.21 | 0.00 | 13.60 | 11.94 |
| Llama→Rebased | 17.41 | 16.63 | 25.27 | 0.00 | 20.48 | 15.96 |
| Llama→Hedgehog | 21.78 | 16.43 | 19.40 | 0.00 | 18.57 | 15.24 |
| *8K Context* | | | | | | |
| SmolLM-360M | 17.84 | 15.44 | 17.29 | 0.17 | 19.06 | 14.16 |
| Llama→xLSTM | 33.71 | 27.66 | 24.86 | 4.25 | 27.61 | 23.62 |
| Llama→GLA | 30.63 | 27.55 | 28.06 | 3.50 | 28.87 | 23.72 |
| Llama→MetaLA | 24.26 | 22.72 | 19.10 | 0.00 | 25.18 | 18.25 |
| Llama→RetNet | 16.70 | 15.85 | 17.25 | 1.50 | 15.05 | 13.27 |
| Llama→DeltaNet | 17.21 | 21.43 | 18.57 | 0.00 | 18.87 | 15.22 |
| Llama→LA | 12.90 | 13.06 | 10.79 | 0.00 | 12.94 | 9.94 |
| Llama→Rebased | 11.98 | 15.61 | 24.65 | 0.50 | 20.66 | 14.68 |
| Llama→Hedgehog | 20.65 | 17.19 | 17.85 | 0.00 | 16.31 | 14.40 |
| *16K Context* | | | | | | |
| SmolLM-360M | 18.12 | 18.01 | 20.29 | 0.00 | 20.96 | 15.47 |
| Llama→xLSTM | 30.31 | 28.19 | 28.25 | 4.00 | 28.77 | 23.90 |
| Llama→GLA | 33.10 | 29.10 | 28.48 | 2.00 | 29.75 | 24.49 |
| Llama→MetaLA | 25.29 | 20.55 | 19.31 | 0.00 | 25.34 | 18.10 |
| Llama→RetNet | 17.16 | 15.89 | 19.90 | 0.00 | 18.19 | 14.23 |
| Llama→DeltaNet | 20.62 | 18.75 | 20.08 | 0.00 | 22.35 | 16.36 |
| Llama→LA | 13.44 | 11.28 | 11.26 | 0.00 | 14.02 | 10.00 |
| Llama→Rebased | 13.21 | 14.81 | 23.64 | 0.00 | 16.25 | 13.58 |
| Llama→Hedgehog | 16.00 | 17.23 | 13.62 | 0.00 | 16.15 | 12.60 |

Table 8: Full evaluation results for long-context evaluation on LongBench benchmark.

# F  ABLATION: SMOLLM-XLSTM COLLECTION

As an outlook, we trained xLSTM student models, based on the SmolLM collection. We used the same training setup as described in Section 4. For the 1.7B model equivalent we also trained a version with a lower learning rate to adjust for size. Results are shown in Table 9.

| MODEL | LAMB. acc. | WINOG. acc. | ARC-E acc. norm. | ARC-C acc. norm. | PIQA acc_norm | HELLAS. acc. norm. | AVG.↑ | RECOVERY |
|---|---|---|---|---|---|---|---|---|
| SmolLM-135M | 32.93 | 52.88 | 55.85 | 29.18 | 68.23 | 42.68 | 46.96 | - |
| SmolLM-360M | 41.33 | 56.51 | 63.72 | 36.01 | 71.49 | 53.37 | 53.73 | - |
| SmolLM-1.7B | 48.38 | 60.93 | 73.48 | 46.42 | 76.06 | 65.74 | 61.83 | - |
| Llama-xLSTM-180M | 26.64 | 50.51 | 51.81 | 26.79 | 67.57 | 39.90 | 43.87 | 93.42% |
| Llama-xLSTM-400M | 35.71 | 56.43 | 60.40 | 32.51 | 70.95 | 50.37 | 51.06 | 95.03% |
| Llama-xLSTM-1.8B | 47.08 | 60.38 | 56.19 | 29.05 | 73.56 | 57.71 | 53.99 | 87.32% |
| Llama-xLSTM-1.8B$_{low-lr}$ | 39.99 | 57.46 | 66.71 | 38.57 | 74.43 | 60.41 | 56.26 | 90.99% |

Table 9: Linearized xLSTM models based on the SmolLM collection. All models were trained with the same 3 Stage regime like in Experiment 1. For the SmolLM-1.7B equivalent, we also trained a version with a lower LR of $1e-4$ for Stage 3.

# G  ABLATION: EFFICIENCY COMPARISON.

Figure 4 shows token generation speed and memory usage across models. Transformer models like Llama incur higher costs due to softmax attention and growing key-value caches. In contrast, linear attention and recurrent models (e.g., xLSTM, GLA) maintain constant or subquadratic memory and achieve faster, linear-time inference through efficient state updates.

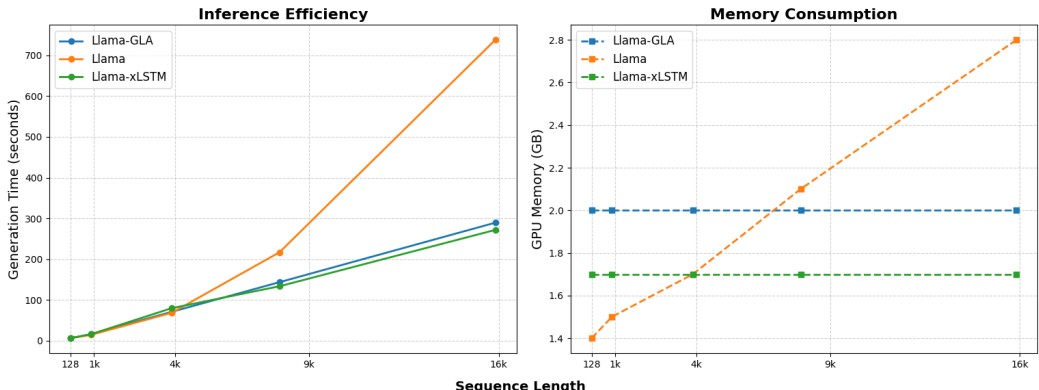

Figure 4: Inference efficiency and memory consumption of linear and softmax attention models, evaluated across single sequences of varying lengths.

