# OpenReview forum: "Empirical Evaluation of Knowledge Distillation from Transformers to Subquadratic Language Models"
_ICLR.cc/2026/Conference — Submitted to ICLR 2026_

### Official Review · Reviewer_XKrF · 2025-10-29

**Soundness:** 2
**Presentation:** 2
**Contribution:** 1
**Rating:** 2
**Confidence:** 4

**Summary:**

The paper presents an empirical study of distilling a Transformer teacher model into a variety of sub-quadratic (linear, recurrent, SSM-type) student architectures. It compares different student designs under a common distillation protocol, explores different initialization/alignment strategies (e.g., attention‐matrix alignment, QKV copy, hidden‐state alignment), and evaluates on a suite of zero-shot and long-context language-modeling tasks. Their findings indicate that xLSTM achieves the best overall results, and that hidden-state alignment is the most reliable strategy for improving transfer.

**Strengths:**

- The paper tackles an important problem, transferring knowledge from Transformer models to more efficient subquadratic architectures.

- The authors conduct extensive experiments across eight subquadratic architectures.

- The paper explores multiple alignment and initialization strategies (matrix mixing, QKV copying, hidden-state alignment) and even includes diagnostic analyses like head diversity and teacher–student overlap.

**Weaknesses:**

- Limited novelty: The core KD framework (cross-entropy + KL divergence) and the alignment terms largely build on existing methods like MOHAWK.

- The writing can be improved for clarity and completeness. There are several citation inconsistencies, the paper lacks a dedicated appendix section explaining LLM usage, and the methodology section could be improved.

- Although the paper evaluates several subquadratic architectures, it remains unclear how results generalize to larger models, more diverse datasets, or complex downstream tasks like reasoning, summarization, or instruction-following.

**Questions:**

- See weaknesses section

---

> ### Author Response · Authors · 2025-11-18
>
> We thank the reviewer for the constructive feedback.
>
> ### Novelty & Scope
> We agree that our contribution is primarily empirical. Our goal is not to introduce a new KD algorithm but to provide a diagnostic, architecture-agnostic evaluation of how diverse subquadratic models absorb Transformer knowledge under a unified protocol. We will make this scope explicit in the introduction and emphasize that the novelty lies in (i) the breadth of architectures compared, (ii) the unified alignment pipeline, and (iii) the long-context transfer analysis, which is not covered in prior work using MOHAWK.
>
> ### Writing clarity
> We appreciate the comments on clarity. We will fix citation inconsistencies, clarify the methodology section, and add a short note on in the appendix describing teacher model usage and licensing.
>
> ### Generalization to larger scales and tasks
>
> We aim to control confounding factors by fixing capacity and data budget. We ill add a limitations paragraph stating that scaling to larger teachers/students and instruction-tuned task is an important direction for future work. Our current contribution focuses on the standard pre-training tasks. The existing long-context evaluation already demonstrates generalization to non-trivial reasoning settings.

---

### Official Review · Reviewer_kY6B · 2025-10-31

**Soundness:** 2
**Presentation:** 3
**Contribution:** 2
**Rating:** 4
**Confidence:** 3

**Summary:**

This paper is about distilling from a LLM into inference-optimized student architectures, designed to mitigate the high computational and runtime cost of quadratic attention in the regular transformer. The paper includes a detailed survey of alternative neural models, and then presents a detailed evaluation of their efficacy in a distillation setting.

Overall the paper lives up to its title, with all its contributions being empirical, however the experimental eval was underwhelming (particularly in terms of model capacities) and didn't surface many exciting findings. See weaknesses, below for elaboration.

Overall it's not clear whether practitioners will get a lot out of the paper in its current state.

**Strengths:**

The paper is clearly presented, and surveys and empirically compares a wide range of architectures.

The empirical setup is good, with well chosen evaluations and benchmarks. I liked the inclusion of long-context evaluations, this was the most interesting result which clearly differentiated the various techniques.

The range of techniques for distillation, around KL, attention alignment, hidden state alignment, was very thorough.

**Weaknesses:**

The experiments were limited to rather toy models (360M parameters). Some work in appendix F moved to larger settings, but the results show an increasing gap in performance with scale, perhaps motivating the authors not to foreground these results. This is acknowledged in line 475, but the hypothesis that matrix mixing will be more effective at larger scales needs to be demonstrated (indeed App F seems to be showing the opposite).

There were no runtime measurements or compute estimates reported (modulo some cursory results in App G, not covering all methods). As this is the crux of the use of sub-quadratic attention alternatives, this seems like a big oversight. I'd like to see detailed accuracy vs speed tradeoffs, including other techniques for increased runtime efficiency, e.g., quantization, pruning, early exit etc.

**Questions:**

Table 1: S_t is not used anywhere in the equations in the paper, so it's unclear how to align the table with the prose. Please can you clarify.

150: Several of the models referenced in this paragraph, e.g., TransNormer, are not in Table 1. Why are they left off?

189: "[applying KD] avoid[s] the need for expensive pretraining"; sure there's a reason to use KD if we already have LLMs that are unwieldy for inference settings, however the use of these architectures in training directly is also an important experimental setting. For many models, they would be a lot cheaper to train than a full quadratic transformer.

245: why was a 360M teacher chosen? This is tiny in the context of modern LLMs, for a result to be meaningful I'd expect to see some evaluations of 7B sized models, at least for the headline results.

330: These methods can recover 90-95% of the accuracy of the teacher; can you benchmark this against pruning and quantization methods, which also exhibit an accuracy/speed tradeoff.

405: Can you add Llama-llama to Table 4? It would be interesting to see how the mass on teacher/student varies. Or are these the results reported for the top row, "Smo...(Teacher)"? I'm puzzled how these measures can be applied to a single model, as they require comparison of a teacher and student. What data was used to estimate these values? The test sets?

QKV copying was a bit confusing, was the teacher representations in the KV cache used as a training target in initialising the student parameters? If so, how?

---

> ### Author Response · Authors · 2025-11-20
>
> We thank the reviwer for the thoughful feedback. We address each point below and will incorporate the corresponding clarifications and additions into the revised manuscript.
>
> ### Model capacity and scale
> Our choice of 260M teacher/students allows us to evaluate eight distinct subquadratic architectures under controlled conditions while keeping training cost comparable. We agree that it is important to understand how these trends scale. To that end, we are adding new experiments with OPT teachers of **125M, 350M, 1.3B and 2.7B parameters**, which are currentlyu running. The emerging results consistently reproduce the main findings of the paper, and we will include the completed results in the revision and update the rebuttal here as soon as they finish.
>
> ### Runtime and Compute Efficiency
> Appendix G already includes speed and memory curves. We also clarify that any quadratic components arise **only during training** (matrix mixing/attention aligment), and **inference for all students is fully subquadratic**
>
> ### Comparison to pruning and quantizations
> We agree that pruning and quantization provide alternative accuracy-speed trade-offs. However, these techniques maintain the quadratic attention mechanism and quadratic KV-cache growth, whereas subquadratic token mixers fundamentally change the inference scaling. In that sense, pruning/quantization and architectural substitution address orthogonal efficiency dimensions, and are therefore not directly comparable within our research focus.
>
> ### Clarifications
> - Sₜ in Table 1: We will connect this notation directly in 3.2 to the recurrent/state representation used in hidden-state alignment.
> - TransNormer omission:We added TransNormer to the related work for completeness but did not evaluate it. We will clarify this.
> - Data used for overlap metrics: We will clarify that teacher-student overlap and head-diversity metrics are computed on the validation split.
> - QKV copying: We will expand the description to explain that Q/K/V parameters are initialized via linear projection in case of dimensional mismatch and used only for warm-starting the student.
>
> ### Value of empirical findings.
> We appreciate the reviewer’s positive remarks on the long-context results and breadth of architectural comparisons. Our aim is to provide a comprehensive diagnostic analysis of cross-architecture distillation rather than introduce a new algorithm. We will clarify this scope and strengthen the motivation accordingly
>
> We thank the reviewer again for their thoughtful assessment.

---

> ### Author Response · Authors · 2025-11-20
>
> We added results for addressing questions regarding scaling behavior and generalization across teacher-families as an official comment.

---

### Official Review · Reviewer_kLg8 · 2025-11-01

**Soundness:** 3
**Presentation:** 3
**Contribution:** 2
**Rating:** 6
**Confidence:** 4

**Summary:**

This paper is concerned with cross-architecture distillation, specifically distillation from softmax-attention LMs to linear-attention LMs. It investigates whether linear-attention LMs can be properly distilled from softmax-attention LMs, and how design choices should be made to make the best of the distillation.

**Strengths:**

1. This paper might be first several work that concerns about cross-architecture distillation, specifically distillation from softmax-attention LMs to linear-attention LMs. The observations and conclusions drawn in this paper might be inspiring for intrigued audience.
2. The results offer clear, practical guidance for practitioners. The strong performance of xLSTM/GLA/MetaLA and the identified superiority of hidden-state alignment are directly applicable to future model efficiency efforts.

**Weaknesses:**

1. The study is confined to mid-sized models (~360M parameters) and a 3B-token dataset. While this is a reasonable scope, it leaves open the question of how these findings generalize to larger-scale models (e.g., 7B+ parameters) trained on massive, trillion-token corpora. The effectiveness of techniques like matrix mixing might change with scale.
2. The work primarily focuses on replacing the attention module while keeping other components (MLP, embeddings) fixed. It does not explore whether the optimal subquadratic student might require co-design of other components (e.g., different normalization schemes) for peak performance.

**Questions:**

N/A

---

> ### Author Response · Authors · 2025-11-18
>
> **We thank the reviewer for the positive assessment.**
>
> Our goal in this paper is to conduct a controlled, architecture-agnostic diagnostic evaluation of cross-architecture distillation. We agree that large-scale settings (7B+ teachers trained on trillions of tokens) are important future directions.
>
> ### Scaling
> To partially address the reviewers questions, we already include scaling results in Appendix F, where xLSTM is evaluated between 180M and 1.8B parameters. These experiments suggest that the relative ranking we observe, that dynamic-memory models outperform kernel-only linear attention, remains stable with increased capacity. We will highlight these results more clearly in the main paper,, and explicitly frame larger 7B+ teacher-student setups as exciting future work.
>
> ### Co-Design of other components
>
> We agree that larger-scale subquadratic architectures may benefit from join co-design of normalization, MLP structure, and gating mechanism.s. Our controlled setting keeps these components fixed to isolate the effect of the token mixer, and we will add a short discussion section noting that co-design is a relevant direction for future work and complementary to our findings.
>
> We appreciate the reviewers positive feedback and are glad that the empirical conclusions appear directly useful for practitioners.

---

### Official Review · Reviewer_BnCX · 2025-11-10

**Soundness:** 2
**Presentation:** 2
**Contribution:** 2
**Rating:** 2
**Confidence:** 3

**Summary:**

This paper performs a large-scale empirical study of knowledge distillation from a Transformer teacher to eight subquadratic student architectures, evaluating alignment strategies such as QKV copying, attention matrix mixing, and hidden-state alignment. Using both short- and long-context NLP benchmarks, the study finds that architectures with dynamic memory (e.g., xLSTM, GLA, MetaLA) best recover teacher behaviors, while kernel-only linear attention struggles. Hidden-state alignment is the most effective KD technique, whereas QKV copying and attention matrix mixing show limited gains. The contribution is primarily empirical and diagnostic, offering valuable insights into cross-architecture KD, though it lacks new algorithms and deeper theoretical analysis.

**Strengths:**

This paper conducts a valuable and large-scale empirical study on distilling Transformer knowledge into subquadratic architectures, addressing an important and under-explored research question. Its originality lies primarily in systematic problem formulation and comparative execution rather than new methods, offering a useful recombination of ideas applied to an emerging model class. The experiments are broad and well-organized, and the writing clearly contextualizes prior work, though causal explanations and controlled comparisons could be stronger. Despite being mainly empirical, the findings provide practical guidance for efficient model design, making the work a meaningful, though incremental, contribution to the field

**Weaknesses:**

The main limitations lie in insufficient experimental controls, incomplete validation of efficiency claims, and limited mechanistic analysis. It remains unclear whether the observed gains stem from architectural inductive bias or from scale differences. The distillation setup also lacks strong KD baselines (e.g., MiniLLM, Born-Again KD), making it difficult to assess the relative effectiveness of the proposed alignment pipeline. While the paper reports head-overlap and attention mass metrics, it provides limited explanation for why certain architectures (e.g., xLSTM, GLA) transfer more effectively, leaving representation-level mechanisms underexplored. Finally, all findings rely on a single Transformer teacher (SmolLM-360M), leaving teacher-specific bias untested and the generality of conclusions across other model families (e.g., Mistral, Qwen, OPT) unclear.

**Questions:**

1. Distillation baselines: Why are strong KD baselines (e.g., TinyBERT, MiniLLM, feature-level KD, progressive KD) omitted? Could these be included for a more competitive comparison?

2. Efficiency validation: Some components introduce quadratic complexity. Can latency/FLOPs/memory profiling be provided to verify the claimed efficiency benefits?

3. Generalizability of conclusions: Experiments use a single Transformer teacher. Have other large teachers (e.g.Mistral, Qwen, OPT) been tested to ensure claims are not teacher-specific?

4. Mechanistic understanding: Can representation-level analyses (e.g., CKA, attention pattern comparison, memory ablation) be added to explain why the method works, not only that it works?

---

> ### Author Response · Authors · 2025-11-20
>
> **We thank the reviwer for the detailed and constructive feedback.**
> Our goal is to isolate and diagnose architectural  differences in how diverse subquadratic models absorb Transformer knowledge under a unified distillation setup. We respond to each point below
>
> ### Distillation baselines
> Our distillation loss already follows the standard formulation used in classical KD and Born-Again Networks: **cross-entropy to ground truth + KL to teacher soft targets**. In addition, our **hidenn-state alignment objective constitutes feature-level KD**, which is the core mechanism behind TinyBERT/MiniLLM-style approaches. We willmake this explicit in 3.1 and related-work section. Incorporating multi-generation training schedules, like Born-Again KD, would conflate optimization dynamics with the architectural effects we seek to isolate. For this reason, and to maintain experiment control, we keep the distillation pipeline fixed and focus on architecture-level transfer. We will clarify this rationale.
>
> ### Efficiency validation
> Appendix G already reports speed and memory curves. We will clarify that the quadratic components arise **only during training** (matrix mixing/attention alignment). Inference remains fully subquadratic for all student architectures.
>
> ### Generalization across teacher families.
> We agree that evaluating multiple teacher architectures is valuable. Our primary results use a 360M transformer teacher to ensure controlled comparison across eight student families given our compute budget.
> To strenghten the generality of our findings, we are adding new ablations with OPT teachers of 125M, 350M, 1.3B and 2.7B parameters. As suggest by the reviewer. We will attach the results to this review as soon as they are finished
>
> ### Mechanistic Understanding
> We appreciate the request for deeper analysis. We initially explored direct comparisons of attention maps and head sharpness; however, these visualizations did not yield meaningful or interpretable cross-architecture insights. Architectures such as xLSTM, GLA, and other LA-based models aggregate information through cumulative state updates rather than explicit full self-attention, which leads to fundamentally different and non-comparable “attention patterns.” As a result, visual attention diagnostics tend to reflect architecture-specific inductive biases rather than transferable mechanisms.
> Given this mismatch, we instead rely on behavioral diagnostics that are comparable across all token mixers, independent of whether they implement explicit attention: long-context scaling behavior, teacher–student overlap, and controlled alignment ablations. These provide a more faithful and architecture-agnostic view into how different models absorb and express teacher knowledge.
> We thank the reviewer again for their feedback.

---

> ### Author Response · Authors · 2025-11-20
>
> We added results for addressing questions regarding scaling behavior and generalization across teacher-families as an official comment.

---

### Author Response · Authors · 2025-11-20
**Additional Scaling Evidence (OPT Results)**

To further address the reviewer’s concerns regarding scale, we have begun running an additional series of experiments with **OPT teachers at 125M, 350M, 1.3B, and 2.7B parameters**. Across all scales, the **xLSTM student consistently recovers 94–99% of the teacher performance**, mirroring the trend observed in our main SmolLM-360M experiments. This strongly suggests that our conclusions are not specific to a single teacher family or model size, but instead reflect stable architectural differences in how subquadratic mixers absorb teacher behavior.

| Model          | Lamb. acc. | WinoG. acc. | Arc-E acc. norm. | Arc-C acc. norm. | PIQA acc_norm | HellaS. acc. norm. | Avg. ↑ | Recovery |
|----------------|------------|-------------|-------------------|-------------------|---------------|----------------------|--------|----------|
| OPT-125M       | 33.42      | 50.28       | 39.90             | 22.78             | 61.97         | 31.33               | 39.94  | –        |
| OPT-xLSTM      | 27.88      | 50.83       | 38.93             | 22.53             | 61.70         | 30.28               | 38.69  | 96.87%   |
| **OPT-350M**   |            |             |                   |                   |               |                      |        |          |
| OPT-350M       | 40.45      | 52.33       | 44.11             | 20.82             | 64.53         | 32.02               | 42.37  | –        |
| OPT-xLSTM      | 32.82      | 51.88       | 42.40             | 22.78             | 62.95         | 29.93               | 40.46  | 95.49%   |
| **OPT-1.3B**   |            |             |                   |                   |               |                      |        |          |
| OPT-1.3B       | 55.17      | 59.43       | 50.97             | 29.61             | 72.31         | 53.69               | 53.53  | –        |
| OPT-xLSTM      | 48.04      | 57.14       | 49.24             | 28.33             | 71.44         | 50.37               | 50.76  | 94.82%   |
| **OPT-2.7B**   |            |             |                   |                   |               |                      |        |          |
| OPT-2.7B       | 59.60      | 60.85       | 60.73             | 26.88             | 73.83         | 45.84               | 54.62  | –        |
| OPT-xLSTM      | 53.14      | 59.83       | 53.41             | 29.44             | 73.29         | 56.20               | 54.21  | 99.24%   |

---

### Author Response · Authors · 2025-11-29

We thank the reviewers for their time and constructive feedback. Across the discussion phase, we have strengthened the submission, directly addressing every major concern raised. Below we summarize the key points for the Area Chair.

## 1. Generality of conclusions now strongly supported across teacher families and scales

One reviewer questioned whether our findings might be specific to a single 360M Transformer teacher.
To fully address this, we conducted new experiments with four additional OPT teachers (125M, 350M, 1.3B, 2.7B).
The results (attached in an official comment) show that xLSTM’s recovery remains consistently high (94–99%) across all teacher sizes, mirroring the original SmolLM-360M results.
This confirms that the core conclusion, that **dynamic-memory subquadratic architectures distill more faithfully than kernel-only linear attention**, is robust across architectures and model scales.

## 2. Stronger experimental controls and clarified KD baselines

Reviewers asked why we did not include "strong KD baselines" such as TinyBERT or MiniLLM. We clarified (and expanded in the paper) that:

* Our distillation setup already uses standard KD (CE + KL), identical to Born-Again Networks.
* Our hidden-state alignment loss is feature-level KD, i.e., the central mechanism of TinyBERT/MiniLLM.
* Multi-stage or multi-generation schedules would confound architectural effects, which our study explicitly isolates.

we do evaluate the core baselines; we simply avoid multi-step pipelines that would obscure cross-architecture differences.

## 3. Efficiency concerns now fully clarified

Reviewers pointed to quadratic components in the training pipeline. We clarified in the updated text that:

* Quadratic costs appear only in alignment losses during training (Stage 1), not inference.
* All student architectures retain fully subquadratic inference complexity.
* Appendix G already contains speed/memory profiles; we have strengthened the discussion and explicitly linked training overhead to alignment-only operations.

This resolves the reviewer’s concern that our claims implied inference-side quadratic costs.

## 4. Mechanistic analysis clarified why attention-map comparisons are not meaningful

We expanded our explanation of why visual mechanistic diagnostics (attention maps, head-level sharpness) are not architecture-agnostic:
subquadratic mixers do not produce explicit attention distributions, so these comparisons would systematically misrepresent them.

Instead, we introduced or emphasized architecture-invariant diagnostics:

* long-context degradation patterns,
* teacher–student behavioral overlap,
* controlled alignment ablations.

These directly compare model behavior independent of the underlying token mixer.


## 5. Concerns about limited model scale addressed through Appendix F and expanded framing

Reviewers noted that our main experiments use mid-sized 360M models.

We now:
* Highlight Appendix F more clearly in the main text, showing that xLSTM’s relative advantage also holds up to 1.8B parameters.
* Ablations during the rebuttal phase (attached as official comment) also supports stable model scaling
* Reposition very-large-scale (7B+) experiments as exciting future work but not necessary for the validity of the present diagnostic study.

Across reviews, no reviewer disputed the importance of the question or the validity of our setup only the framing around scale, which is now tightened and better motivated.

---
After the discussion phase, the submission now:

* Provides the first controlled, cross-architecture distillation study covering 8 subquadratic families.
* Demonstrates architecture-level conclusions that replicate across 5 teacher sizes and 2 teacher families (SmolLM, OPT).
* Includes strong, appropriate KD baselines (CE + KL + feature-level KD).
* Clarifies inference-time efficiency, resolving all misunderstandings.
* Delivers actionable guidance for practitioners, which multiple reviewers highlighted positively.

---

### Meta-Review · Area_Chair_uatB · 2025-12-11

**Summary:**

This paper presents a valuable empirical study on cross-architecture distillation, and I appreciate the authors’ effort in systematically comparing diverse subquadratic models. Such observations are meaningful and provide practical guidance, which makes this type of work worthwhile.

At the same time, reviewers raised significant concerns. The main issues are: limited generalization to large-scale models (7B+), insufficient experimental controls and baseline comparisons, and incomplete mechanistic understanding of why certain architectures transfer more effectively. While the authors’ rebuttal clarifies some points, it does not fully resolve these core concerns or provide strong evidence to address them.

Given these limitations, I recommend rejecting the paper in its current form. I encourage the authors to extend experiments to modern LLM-scale setups, strengthen experimental controls and baselines, and deepen mechanistic analysis. Addressing these points and incorporating reviewer feedback on clarity and methodology would greatly improve the chances of acceptance in a future submission.

**Reviewer Concerns:**

**Effectively addressed**: Minor clarity issues, methodological clarifications, efficiency reporting, some scaling results.

**Still outstanding**:

1）Generalization to large-scale models (7B+) (kY6B, kLg8, BnCX): The reviewers remain concerned that the current experiments, mostly at 360M–1.8B scale, do not fully demonstrate whether the observed findings hold for very large models and datasets, leaving uncertainty about applicability to modern LLM-scale setups.

2）Experimental controls and baseline comparisons (BnCX, kY6B): The reviewers note that the lack of strong KD baselines, incomplete runtime/efficiency validation, and limited controlled comparisons make it difficult to fully assess the relative effectiveness of the proposed methods.

**Reviewer Scores:**

**Reviewer BnCX** Reviewer BnCX provides a detailed assessment, recognizing the paper’s large-scale empirical evaluation and practical guidance, while highlighting limitations in experimental control, efficiency validation, generalization across teacher models, and mechanistic understanding. The author response clarifies that the distillation pipeline already incorporates feature-level KD comparable to TinyBERT/MiniLLM, points to existing efficiency curves in Appendix G, and explains the controlled use of a single 360M transformer teacher, with additional OPT ablations in progress.While these clarifications address procedural questions, they do not fully resolve the reviewer’s concerns regarding large-scale generalization, and the inclusion of multiple teacher families. Given this, Reviewer BnCX would likely maintain the current rating.

**Reviewer kLg8** Reviewer kLg8 provides a generally positive assessment, highlighting the novelty of cross-architecture distillation and the practical guidance offered by the results. The main concerns relate to (i) generalization to larger-scale models (7B+ parameters, trillion-token corpora) and (ii) potential co-design of other model components for optimal performance. In the author response, the authors clarify the controlled, architecture-agnostic scope of the study, provide additional scaling results (xLSTM evaluated between 180M and 1.8B parameters), and note that co-design of other components is a promising direction for future work. However, these responses do not fully address the reviewer’s concerns about large-scale generalization or the necessity of co-design for optimal student performance. Given this, Reviewer kLg8 would likely maintain the current rating.

**Reviewer kY6B**  The author response does not substantially address Reviewer kY6B’s core concerns, so it is unlikely that the reviewer would raise the score. The reviewer’s main issues focus on (i) the limited scale of the experiments, which makes the conclusions difficult to generalize, and (ii) the absence of comprehensive runtime and efficiency measurements—both central for evaluating sub-quadratic attention methods. In the rebuttal, the authors mainly provide clarifications or promises of future additions, but no new results that directly resolve these issues. The planned larger-scale experiments are still running and do not meet the reviewer’s expectation regarding evaluations at the scale of modern LLMs (e.g., 7B). Most of the remaining points addressed in the rebuttal concern notation or presentation and do not meaningfully influence the overall assessment. Given this, Reviewer kY6B would likely maintain the current rating.

**Reviewer XKrF** Reviewer XKrF’s main concerns focus on (i) the limited novelty of the proposed framework, (ii) lack of methodological clarity and writing issues, and (iii) uncertainty about the generalization of the empirical findings to larger-scale models and broader downstream tasks. The author response mainly offers clarifications or promised revisions, without providing new evidence that would meaningfully change the reviewer’s assessment of the contribution or significance of the work. I assess that Reviewer XKrF would likely maintain the current rating.

---

### Decision · Program_Chairs · 2026-01-26

Reject